# Relationship between Maternal Socioeconomic Factors and Preterm Birth in Latvia

**DOI:** 10.3390/medicina60050826

**Published:** 2024-05-17

**Authors:** Katrīne Kūkoja, Anita Villeruša, Irisa Zīle-Velika

**Affiliations:** 1Institute of Social, Economic and Humanities Research, Vidzeme University of Applied Sciences, LV-4201 Valmiera, Latvia; 2Department of Public Health and Epidemiology, Rīga Stradiņš University, LV-1010 Riga, Latvia; anita.villerusa@rsu.lv; 3Department of Research and Statistics, Centre for Disease Prevention and Control, LV-1005 Riga, Latvia; irisa.zile@spkc.gov.lv

**Keywords:** preterm birth, socioeconomic factors, mothers’ education, mother’s nationality

## Abstract

*Background and Objectives:* Worldwide, preterm birth (PTB) stands as the primary cause of mortality among children under 5 years old. Socioeconomic factors significantly impact pregnancy outcomes, influencing both maternal well-being and newborn health. Understanding and addressing these socioeconomic factors is essential for developing effective public health interventions and policies aimed at improving pregnancy outcomes. This study aims to analyse the relationship between socioeconomic factors (education level, marital status, place of residence and nationality) and PTB in Latvia, considering mother’s health habits, health status, and pregnancy process. *Materials and Methods:* A cross-sectional study was conducted using data from the Medical Birth Register (MBR) of Latvia about women with singleton pregnancies in 2022 (*n* = 15,431). Data analysis, involving crosstabs, chi-square tests, and multivariable binary logistic regression, was performed. Adjusted Odds ratios (aOR) with 95% confidence intervals (CI) were estimated. *Results:* Lower maternal education was statistically significantly associated with increased odds of PTB. Mothers with education levels below secondary education had over two times higher odds of PTB (aOR = 2.07, *p* < 0.001, CI 1.58–2.70) and those with secondary or vocational secondary education had one and a half times higher odds (aOR = 1.58, *p* < 0.001, CI 1.33–1.87) after adjusting for other risk factors. Study results also showed the cumulative effect of socioeconomic risk factors on PTB. Additionally, mothers facing two or three socioeconomic risk factors in Latvia exhibited one and a half times higher odds of PTB (aOR = 1.59, *p* = 0.021). *Conclusions:* The study highlights the cumulative impact of socioeconomic risk factors on PTB, with higher maternal education demonstrating the highest protective effect against it. This underscores the importance of education in promoting optimal foetal development. Since the influence of socioeconomic factors on PTB is not a widely studied issue in Latvia, further research is needed to improve understanding of this complex topic.

## 1. Introduction

Preterm birth (PTB), complications during childbirth, infections, and congenital defects are the primary reasons for neonatal mortality. Reducing the global impact of PTB is crucial to lowering neonatal and child mortality, morbidity rates and meeting the Sustainable Development Goal target 3.2. This target aims to decrease neonatal mortality to 12 or fewer neonatal deaths per 1000 live births in all countries [1]. According to the World Health Organization’s data, more than 10% of all infants are born prematurely each year and globally prematurity is the leading cause of death in children under the age of 5 years. In addition, unfavourable pregnancy outcomes have serious consequences for an individual’s health throughout the life-course. PTB elevates the risk of childhood neurologic disability, respiratory diseases, gastrointestinal disorders, cardiovascular disease, and compromises immunity. Ill health at birth is also an important predictor of long-term quality of life, increasing the likelihood of disability, poorer education, employment, income, and so on [2,3,4,5,6,7,8]. In Latvia, there has been a slightly upward trend in the PTB rate over the past five years. The rate increased from 5.3 PTB per 100 births in 2017 to 5.8 PTB per 100 births in 2022 [9].

Genetic and lifestyle factors are known to have a significant impact on maternal health and pregnancy outcomes. However, it is important to highlight that social determinants of health also play a crucial role in shaping these outcomes and remain one of the most reliable predictors of health disparities, increasingly used in studying pregnancy outcomes [10,11,12,13,14,15,16]. The primary employed individual-level determinants of socioeconomic status (SES) are education level, income, and employment status. Additional determinants encompass neighbourhood income, family structure, race/ethnicity, place of residence, and wealth. These socioeconomic factors have been included as variables in several previous studies that have demonstrated an association of poorer pregnancy outcomes with lower SES [17,18,19,20,21,22,23,24]. The links between SES and adverse pregnancy outcomes are observed even in prosperous countries with a universal healthcare system suggesting that the associations observed may not be attributable to the type of prenatal care received [12,25,26,27].

Latvia can be characterized as a state experiencing a declining population trend over the past few decades. If in 1990 the population stood at 2.66 million, by 2023, it had decreased to 1.88 million [28]. Additionally, statistical data from 2022 reveal that 21.6% of individuals (22.7% females) residing in the European Union (EU-27) were at risk of poverty or social exclusion. In Latvia, this figure was notably higher, standing at 26%, sharing the fourth position with Spain (26%). This ranking positions Latvia among the countries with the highest risk of poverty and social exclusion within the EU-27, trailing behind Romania, Bulgaria, and Greece. Notably, the percentage of women at risk of poverty and social exclusion is even higher, reaching 28.8% [29].

Considering that nearly one-third of women in Latvia face the risk of poverty or social exclusion and given the limited research on the association between maternal socioeconomic factors and PTB in Latvia, it is important to explore this topic in further detail.

## 2. Materials and Methods

A cross-sectional study was carried out using data from the Medical Birth Register (MBR) of Latvia, which is based on the standardised birth information submitted by maternity units across the country [30].

The study was restricted to all women with singleton pregnancies in Latvia during 2022. Multiple pregnancies were intentionally excluded from the study to enhance the precision of the analysis and reduce confounding variables related to PTB. Additionally, mothers residing abroad (*n* = 12) and those with post-term delivery (>42 weeks) were excluded from the study (*n* = 32).

In total, data on 15,431 singleton births were included in the study out of the original sample (*n* = 15,475). Gestational age was classified as either PTB (22–36 weeks) or term birth ((TB) 37–42 weeks). The age range of mothers spanned from 13 to 54 years old, with an average maternal age of 30.8 years (SD 5.6). Among all mothers included in the sample, 7.5% (95% CI, 7.1–8.0%) reported smoking and/or using alcohol and/or narcotic substances during pregnancy. Before pregnancy, 35.3% (95% CI, 34.5–36.1%) were overweight or obese, and 5% (95% CI, 4.7–5.4%) were underweight. Most mothers experienced at least one maternal comorbidity (51.6%, 95% CI, 50.8–52.4%), received complete antenatal care by attending all required visits (94.6%, 95% CI, 94.2–94.9%), and had no pregnancy complications (62.1%, 95% CI, 61.3–62.9%). A total of 61.2% (95% CI 60.4–61.9%) of births included in the sample were multiparous. Please see Table 1 about baseline socioeconomic characteristics of the study participants.

The following socioeconomic characteristics of mothers were considered as independent variables in the study, based on previous studies [14,17,26,27]: maternal education level (lower than secondary education (ISCED 0–2)/secondary or vocational secondary education (ISCED 3–4)/higher education (ISCED 5–8)) [31], marital status (single/not single (marriage or cohabitation)), place of residence (capital city/regional city/rural region), and nationality (Latvian/other nationality).

Socioeconomic factors were adjusted for maternal age (<=19; 20–34, >=35), maternal unhealthy habits (smoking and/or alcohol use and/or narcotic substances use—yes/no), mother’s body mass index (BMI) before pregnancy (normal (18.5–25), underweight (<18.5), overweight or obesity (=>25)) [32], maternal comorbidities (yes/no), antenatal care usage (complete (the woman registered her pregnancy up to the 12th week and received antenatal monitoring in compliance with national regulations)/incomplete or non-existent (the woman did not receive adequate antenatal care as per national regulations and/or did not register her pregnancy) [33], parity (primipara/multiparous) and pregnancy complications (yes/no).

The data were analysed using IBM SPSS Statistics 29 and crosstabs, chi-square tests, and logistic regression. The odds ratio (OR) and 95% confidence intervals (CI) were calculated with a significance level of *p* < 0.05 to determine associations between PTB and mothers’ socioeconomic characteristics. Continuous variables were summarized using the mean and standard deviation, while categorical variables were presented as frequencies and percentages in descriptive statistics.

Logistic regression models were utilized to explore socioeconomic differences in PTB, with higher education, Latvian nationality, living in the capital city, and mothers who are not single defined as the reference groups based on the literature review.

Initially, individual binary logistic regression models assessed the association between each socioeconomic variable and PTB. Then, five multivariable binary logistic regression models were created. The first model included all significant socioeconomic factors, the second adjusted these factors among themselves and included maternal characteristics by age, ill health status and unhealthy habits, and the third added pregnancy-related characteristics (pregnancy complications and antenatal care usage). Variables were systematically added to the model one at a time to observe their individual effects and discern any differences.

The impact of exposure to one or two to three socioeconomic risk factors (less than secondary education, nationality other than Latvian, and being single) was compared to no exposure (reference value), adjusting the odds also by maternal age, ill health status and unhealthy habits in the fourth model. In the fifth model odds were additionally adjusted by pregnancy-related characteristics.

## 3. Results

Out of the 15,431 singleton births that were included in the study, there were 802 PTB (5.2% (95% CI 4.9–5.6%)). Table 2 reveals statistically significant differences in the distribution of PTB and TB across various maternal education levels, as well as among mothers with nationalities other than Latvian. There was higher prevalence of mothers with education levels lower than secondary (16.0%, 95% CI, 13.5–18.7%) and secondary/vocational secondary education (45.8%, 95% CI, 42.3–49.3%) among PTB cases compared to TB. Conversely, mothers with higher education are more prominently represented in TB cases, accounting for 52.9% (95% CI, 52.1–53.7%) of the total. Data also showed a 4.6% higher rate of PTB (26.8% (95% CI, 23.8–30.0%) among mothers with other nationalities compared to TB (22.2% (95% CI, 21.6–22.9%). No statistically significant differences in PTB and TB distribution were observed regarding marital status and place of residence (Table 2).

Additionally, statistically significant differences were observed in the distribution of PTB and TB across maternal BMI before pregnancy (except for obesity and overweight category), maternal comorbidities, incomplete or non-existent antenatal care utilization, and unhealthy habits during pregnancy and pregnancy complications. No statistically significant association was found regarding parity (Table 2).

The initial regression analysis assessing the odds of experiencing PTB revealed statistically significant findings. Mothers with higher education displayed significantly lower crude OR of having PTB compared to those with less than secondary education (OR = 2.44, *p* < 0.001) and secondary or vocational secondary education (OR = 1.67, *p* < 0.001). Additionally, mothers of other nationalities exhibited higher odds of PTB compared to Latvian mothers (OR = 1.28, *p* = 0.003).

However, being married or in a relationship showed only borderline statistically significant protective effects against PTB (crude OR = 1.58, *p* = 0.105) compared to single mothers. And residing in a capital city did not demonstrate statistically significant protection against PTB odds in Latvia compared to regional cities (OR = 1.09, *p* = 0.386) or rural areas (OR = 1.06, *p* = 0.514). As a result, place of residence was excluded from subsequent regression models developed in this study.

Nearly all analysed factors related to maternal health, health behaviour, and the pregnancy process exhibited statistically significant crude odds of PTB, except for parity and mothers’ BMI before pregnancy (overweight or obese vs. normal weight) (Figure 1).

### 3.1. The Relationship between Socioeconomic Factors and PTB in Three Multivariate Logistic Regression Models

Within this study, three multivariable logistic regression models were developed to identify which of the analysed socioeconomic factors have an impact on PTB in Latvia.

The analysis showed that even after adjusting for other risk factors, higher maternal education serves as a protective factor against PTB. The odds of PTB for mothers with education below secondary level compared to mothers with higher education slightly decrease after adjusting for marital status and nationality (aOR = 2.41, *p* < 0.001). They further decrease when risk factors related to maternal health and health behaviour, such as maternal comorbidities, maternal BMI before pregnancy, unhealthy habits of the mother, and maternal age were added to the model (aOR = 2.18, *p* < 0.001). Additionally, adjusting for risk factors related to the pregnancy process, such as antenatal care usage and pregnancy complications, also leads to a decrease in the odds of PTB (aOR = 2.07, *p* < 0.001).

The same trend with slightly lower odds was observed in the case of secondary or vocational secondary education, with OR slightly declining but remaining statistically significant after adjusting for other socioeconomic factors (aOR = 1.65, *p* < 0.001), risk factors related to maternal health and health behaviour (aOR = 1.61, *p* < 0.001), and pregnancy process (aOR = 1.58, *p* < 0.001) (Table 3).

A deeper analysis reveals that while the described association between education and PTB slightly strengthened with the inclusion of maternal age and maternal comorbidities in the model, the inclusion of unhealthy habits, maternal BMI before pregnancy, and antenatal care usage weakened this association. This can be attributed to a statistically significant association (*p* < 0.001) between mothers’ BMI before pregnancy and education level, and with higher-educated mothers being more likely to have a normal weight. There was also a notable decrease in unhealthy habits (*p* < 0.001) and higher antenatal care usage among higher-educated mothers (*p* < 0.001). Specifically, 18.7% (*n* = 272) of women with lower education levels received incomplete or no antenatal care, compared to 6.7% (*n* = 398) of those with secondary education and 2.1% (*n* = 166) of those with higher education.

Regression analysis revealed that mothers with other nationalities had higher odds of PTB. This relationship remained after adjusting for other socioeconomic factors (aOR = 1.25, *p* = 0.008). However, after adjusting for factors that characterize the mother’s ill health and unhealthy lifestyle habits (aOR 1.17, *p* = 0.073) and factors that characterize the process and course of pregnancy, the odds of PTB decreased (aOR = 1.16, *p* = 0.085). Although this factor lost statistical significance at the 95% CI, its proximity to the significance threshold justifies the inclusion of this factor in all three regression models (Table 3).

The decrease in the association between nationality and PTB may be attributed to differences in the prevalence of other factors. More Latvian mothers avoided unhealthy habits (93.1%, *n* = 11,138) compared to mothers of other nationalities (90.3%, *n* = 3132) (*p* < 0.001), and non-Latvian mothers had a slightly higher mean age at delivery (mean = 31.5, SD 5.8) compared to Latvian mothers (mean = 30.6, SD 5.6 years). Additionally, 7.3% of non-Latvian mothers had incomplete or no antenatal care, compared to 4.9% of Latvian mothers (*p* < 0.001).

Conversely, the association between marital status and PTB already weakens upon adjustment for maternal education level and nationality, indicating diminished odds for single mothers to have PTB (aOR = 1.18, *p* = 0.572) and a rise in the *p*-value. This led to the exclusion of marital status from the second and third model. This trend can be explained by the fact that fewer mothers in the sample with higher education were single (0.5%) compared to those with secondary or vocational secondary education (1.1%) and mothers with education below secondary level (4.9%) (*p* < 0.001). Additionally, mothers of non-Latvian nationality (1.7%) were more likely to be single than Latvian mothers (1%) (*p* < 0.001).

### 3.2. Multivariate Logistic Regression Models for PTB on Combinations of Socioeconomic Risk Factors

As highlighted previously, unadjusted data analysis identified three statistically significant socioeconomic risk factors for PTB in Latvia: (1) “less than secondary education”, (2) “other nationality than Latvian” and (3) “being single”. These three factors were included in the fourth model and fifth model. Among the mothers included in the sample, 69.8% (95% CI, 69.1–70.6%) had none of the risk factors mentioned before (*n* = 10,777), 27.4% (95% CI, 26.7–28.1%) had one risk factor (*n* = 4234) and 2.7% (95% CI, 2.5–3.0%) had two or three risk factors (*n* = 420).

Regression analysis shows that the presence of any (one) of three risk factors increased the odds of having PTB compared to none, even after adjusting the odds for maternal comorbidities, maternal BMI before pregnancy, unhealthy habits of the mother, and maternal age (aOR = 1.34, *p* < 0.001), as well as pregnancy complications, and antenatal care usage (aOR =1.31, *p* = 0.001). While mothers who had two or three socioeconomic risk factors had even higher odds of PTB in Latvia, even after adjusting for factors related to maternal health and heath behaviour (aOR = 1.70, *p* = 0.007), as well as factors related to the pregnancy process, which intensified the odds of PTB by over one and a half times (aOR = 1.59, *p* = 0.021) (Table 4).

## 4. Discussion

Approximately 500,000 infants are born prematurely in Europe annually, and this figure is rising [34]. Also, statistics about the situation in Latvia show that the percentage of PTB is rising, with 802 infants born preterm in 2022 in singleton pregnancies.

Consistent with prior research conducted in other countries [17,21,25,35], this study underscores the association between socioeconomic factors and PTB, revealing two main statistically significant socioeconomic factors influencing PTB in Latvia: education level and nationality. Additionally, this study showed the cumulative effects of socioeconomic risk factors on PTB.

Higher maternal education exhibited a protective effect, with the likelihood of PTB decreasing with increased education levels. Education level was found to be the most impactful socioeconomic determinant of PTB in Latvia. This underscores the importance of education in fostering optimal foetal development, a finding consistent with existing literature [26]. Similar results were observed in a study conducted in Abu Dhabi, UAE, which associated lower levels of maternal education, specifically below secondary level, with an increased risk of PTB [15]. Additionally, a study from Denmark highlighted maternal educational level as the most significant socioeconomic indicator, among maternal and paternal education level, occupation, and household income, influencing PTB [35].

The study findings revealed various factors that influence the association between maternal education and PTB in Latvia. Mothers with lower educational attainment were more prone to engaging in unhealthy habits during pregnancy, lacking complete or any antenatal care, experiencing being underweight, overweight, or obese, as well as facing maternal comorbidities and pregnancy complications. These findings suggest that education level influences PTB through health literacy, healthcare utilization, and health behaviours, etc. It is essential to recognize that numerous health issues impacting pregnant women and children often arise from risky and unhealthy behaviours, which can be mitigated through increased awareness and knowledge dissemination.

Until now Latvian policy planning documents, such as the Maternal and Child Health Plan 2018–2020 [36] and the Public Health Guidelines 2021–2027 [37] have primarily emphasized health system enhancements and strategies within the healthcare sector in promoting the health of mothers and children, overlooking the significance of socioeconomic determinants. Given the close connection between education level and health literacy, an area that is understudied in Latvia, educational initiatives should also specifically target less educated groups of society.

Considering that not all mothers will achieve higher education levels, integrating discussions about pregnancy, its risk factors, and the significance of antenatal care usage into the secondary education curriculum could potentially enhance birth outcomes. Moreover, leveraging social media platforms for educating society about these topics can yield significant benefits. Nonetheless, additional research is needed to assess the effectiveness of these and other optional educational strategies, including parental education programmes targeted at less educated groups of society, etc.

When it comes to nationality, prior research conducted across various countries has consistently demonstrated that maternal nationality influences the occurrence of PTB. Results of a study conducted in Japan showed that nationality had a statistically significant impact on the relative risk of PTB, with Filipino and Brazilian mothers showing the highest risks, while Korean and Chinese mothers had lower relative risks compared to Japanese mothers [23]. Another study in England and Wales spanning from 2006 to 2012 also revealed ethnic disparities in PTB rates. The results showed that babies from Black Caribbean, Indian, Bangladeshi, Pakistani, and Black African backgrounds had significantly increased odds of PTB compared to White British babies (ORs ranging between 1.04–1.25) [20].

In Latvia, initial findings suggested an association between maternal nationality and PTB rates. Even though after adjusting for other risk factors association decreased, it persisted (aOR = 1.16; *p* = 0.085) at a level consistent with studies conducted in other countries. Further analysis revealed that this decrease was largely influenced by maternal BMI, unhealthy habits, maternal age, and antenatal care usage, emphasizing the importance of considering multiple factors when investigating PTB rates among different ethnic groups. A more in-depth investigation into this pattern is needed in the Latvian context.

Marital status showed no significant relationship with PTB in Latvia after adjusting for education and nationality (aOR = 1.18, *p* =0.572). Similar results were obtained in the Maher et al. (2023) study, which examined the association between individual-level socioeconomic factors (including education, employment, relationship status, income) and adverse pregnancy and neonatal outcomes (including PTB) using data from Sweden, Netherlands, and the Republic of Ireland. Although the data showed that single or divorced mothers had 1.38 higher odds of experiencing PTB, the association was not statistically significant (CI 0.70–2.71) [21].

Contrary to the results of previous studies conducted in other countries, which have indicated that women who live in rural and remote areas are more likely to experience PTB compared to their urban and city counterparts [38], place of residence demonstrated no substantial effect on PTB rates in Latvia. Possibly it is due to comprehensive state-funded antenatal care accessible to all pregnant women, encompassing doctor and midwife consultations, and laboratory and diagnostic tests at specific intervals during pregnancy. Enhanced access to antenatal care likely contributes to improved pregnancy outcomes, potentially mitigating the impact of regional disparities on the occurrence of PTB in Latvia [39]. The lower level of urbanization in Latvia and the compact size of the country may also play a role, particularly considering that regional cities have relatively small populations and that travel distances to antenatal care visits are typically short, potentially minimizing barriers to access.

Additionally, study results underscored the cumulative impact of socioeconomic risk factors on PTB, with mothers exposed to multiple risk factors facing elevated odds of PTB (aOR = 1.59, *p* = 0.021). This finding emphasizes the importance of addressing multiple socioeconomic determinants in maternal and neonatal health. It aligns with similar research conducted in Sweden, the Netherlands, and Ireland, which revealed that mothers exposed to ≥two risk factors such as “less than a third level of education”, “not in paid employment”, “single/separated/divorced”, and “low level of income” were more than one and a half times as likely to experience PTB (aOR: 1.75, 95% CI: 1.06–2.89) [21].

Given that findings of this study align closely with those of other countries with universal antenatal care systems, we suggest that these results may be applicable to other developed countries with similar comprehensive antenatal care coverage [12,17,21].

### Strengths and Limitations

Due to the unavailability of individual-level data on income and occupation in the MBR of Latvia and the inability to integrate the MBR with other databases to access these data, our analysis was limited to utilizing education level and alternative socioeconomic indicators such as marital status, place of residence, and nationality. While these indicators provide insight into SES, they may not fully capture the nuances of income and occupation.

However, it is worth noting that in countries with universal antenatal care coverage, most studies exploring the impact of individual-level, household, or neighbourhood-level income on birth outcomes, including PTB, have not found statistically significant associations [12,17,21]. This lack of association could be attributed to the safety net provided for mothers and children of lower SES within these systems. Conversely, in countries like the United States of America, where comprehensive support systems are lacking, the correlation between SES and birth outcomes tends to be stronger [13,22]. This suggests that even if included in the study, maternal income may not have a significant role in PTB in Latvia. Nonetheless, further research is necessary for a more comprehensive understanding of this relationship within the Latvian context.

This study also boasts several notable strengths. As previously mentioned, it leveraged a substantial dataset from the MBR of Latvia. The complete coverage of birth registration data on singleton births, the large number of births, and the low proportion with missing data minimize selection bias, detection bias, and observer bias. Secondly, the study employed a comprehensive set of individual-level socioeconomic factors to evaluate SES. This nuanced approach enhances our ability to discern the direct influence of SES on PTB, offering a more insightful perspective than conventional aggregate area-based measures.

This study sheds light on the relationship between socioeconomic factors and PTB in Latvia, filling a gap in the existing literature on socioeconomic factors as risk factors for PTB. Study results emphasize the need for holistic approaches to maternal and neonatal health that address not only medical risks, lifestyle, and use of health care services, but also socioeconomic determinants.

## 5. Conclusions

In Latvia, mothers’ education emerges as the most significantly associated socioeconomic factor with PTB, followed by mothers’ nationality, while no significant associations were found for place of residency and marital status. Mothers with lower education levels have significantly elevated odds of PTB compared to those with higher education, highlighting the importance of targeted public health interventions aimed at improving pregnancy outcomes for mothers in lower educational groups. In addition, supporting mothers from non-Latvian backgrounds, particularly by enhancing access to antenatal care and addressing unhealthy habits, should be considered for preventing PTB in Latvia.

The study results highlighted the cumulative effect of multiple socioeconomic risk factors on PTB rates in Latvia. Results show that mothers exposed to two or three risk factors face increased odds of PTB and emphasize the need for a comprehensive approach to prenatal care.

## Figures and Tables

**Figure 1 medicina-60-00826-f001:**
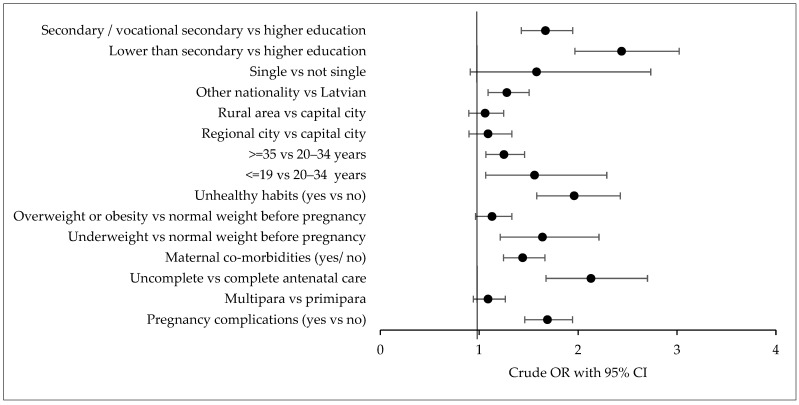
Maternal socioeconomical and prenatal factors associated with PTB (crude OR). OR (odds ratio), CI (confidence interval).

**Table 1 medicina-60-00826-t001:** Maternal baseline socioeconomic characteristics.

Variables	Total (*n* = 15,431)	
	N	% (95% CI)
Mothers’ education level		
Higher education	8051	52.2 (51.4–53.0)
Secondary/Vocational secondary	5928	38.4 (37.6–39.2)
Lower than secondary	1452	9.4 (9.0–9.9)
Marital status		
Married	9685	62.8 (62.0–63.5)
Cohabitation	5569	36.1 (35.3–36.9)
Single	177	1.1 (1.0–1.3)
Nationality		
Latvian	11,963	77.5 (76.9–78.2)
Russian	2043	13.2 (12.7–13.8)
Ukrainian	261	1.7 (1.5–1.9)
Polish	142	0.9 (0.8–1.1)
Belarusian	111	0.7 (0.6–0.9)
Lithuanian	70	0.5 (0.4–0.6)
Other nationalities	841	5.5 (5.1–5.8)
Place of residency		
Capital city	4880	31.6 (30.9–32.4)
Regional city	3346	21.7 (21.0–22.3)
Rural area	7205	46.7 (45.9–47.5)

N (total number of births), CI (confidence interval). All reported percentages are valid percentages after considering missing data at each variable level.

**Table 2 medicina-60-00826-t002:** Maternal baseline socioeconomic characteristics and association between the proportion of PTB and TB.

Variables	Gestation Age at Delivery				
	TB (*n* = 14,629)		PTB (*n* = 802)		*p*-Value *
	N	% (95% CI)	N	% (95% CI)	
Mothers’ education level					
Higher education	7744	52.9 (52.1–53.7)	307	38.3 (34.9–41.7)	<0.001
Secondary/Vocational secondary	5561	38.0 (37.2–38.8)	367	45.8 (42.3–49.3)	<0.001
Lower than secondary	1324	9.1 (8.6–9.5)	128	16.0 (13.5–18.7)	<0.001
Marital status					
Not single	14,466	98.9 (98.7–99.0)	788	98.3 (97.1–99)	0.849
Single	163	1.1 (1.0–1.3)	14	1.7 (1.0–2.9)	0.104
Nationality					
Latvian	11,376	77.8 (77.1–78.4)	587	73.2 (70.0–76.2)	0.149
Other nationality	3253	22.2 (21.6–22.9)	215	26.8 (23.8–30.0)	0.008
Place of residency					
Capital city	4637	31.7 (30.9–32.5)	243	30.3 (27.1–33.6)	0.488
Regional city	3165	21.6 (21.0–22.3)	181	22.6 (19.7–25.6)	0.585
Rural area	6827	46.7 (45.9–47.5)	378	47.1 (43.6–50.7)	0.859
Mothers BMI before pregnancy					
Normal weight	8039	59.8 (59.0–60.7)	387	55.5 (51.7–59.3)	0.012
Underweight	660	4.9 (4.6–5.3)	52	7.5 (5.6–9.7)	0.011
Overweight or obesity	4735	35.2 (34.4–36.1)	258	37.0 (33.4–40.7)	0.917
Unhealthy habits					
Yes	1055	7.2 (6.8–7.6)	106	13.2 (10.9–15.8)	<0.001
No	13,574	92.8 (92.4–93.2)	696	86.8 (84.2–89.1)	0.083
Maternal comorbidities					
No	7148	48.9 (48.0–49.7)	320	39.9 (36.5–43.4)	<0.001
Yes	7481	51.1 (50.3–52.0)	482	60.1 (56.6–63.5)	<0.001
Antenatal care usage					
Complete	13,876	94.9 (94.5–95.2)	719	89.7 (87.3–91.7)	0.136
Incomplete or non-existent	753	5.1 (4.8–5.5)	83	10.3 (8.3–12.7)	<0.001
Parity					
Primipara	5697	38.9 (38.2–39.7)	296	36.9 (33.6–40.4)	0.363
Multipara	8932	61.1 (60.3–61.8)	506	63.1 (59.6–66.4)	0.480
Pregnancy complications					
No	9180	62.8 (62.0–63.5)	401	50.0 (46.5–53.5)	<0.001
Yes	5449	37.2 (36.5–38.0)	401	50.0 (46.5–53.5)	<0.001

TB (term birth), PTB (preterm birth), N (total number of births), CI (confidence interval), BMI (Body mass index). All reported percentages are valid percentages after considering missing data at each variable level. * chi-square test is used.

**Table 3 medicina-60-00826-t003:** Adjusted association between socioeconomic factors and PTB during 2022 in Latvia—multivariable binary logistic regression models.

Variables	Model No. 1		Model No. 2		Model No. 3	
	aOR (95% CI) ^1^	*p*-Value	aOR (95% CI) ^2^	*p*-Value	aOR (95% CI) ^3^	*p*-Value
Mothers’ education						
Higher	Reference		Reference		Reference	
Lower than secondary	2.41 (1.93–2.99)	<0.001	2.18 (1.68–2.85)	<0.001	2.07 (1.58–2.70)	<0.001
Secondary/Vocational secondary	1.65 (1.41–1.93)	<0.001	1.61 (1.36–1.91)	<0.001	1.58 (1.33–1.87)	<0.001
Nationality						
Latvian	Reference		Reference		Reference	
Other nationality	1.25 (1.06–1.46)	0.008	1.17 (0.99–1.39)	0.073	1.16 (0.98–1.38)	0.085
Marital status *						
Not Single	Reference					
Single	1.18 (0.67–2.05)	0.572	NA		NA	

^1^ aOR for mothers’ education level, nationality, and marital status. ^2^ aOR for mothers’ education level, nationality, maternal comorbidities, maternal BMI before pregnancy, unhealthy habits of the mother and maternal age. ^3^ aOR for mother’s education level, nationality, maternal comorbidities, maternal BMI before pregnancy, unhealthy habits of the mother, maternal age, pregnancy complications, and antenatal care usage. * Marital status was not included in the 2nd and 3rd model. aOR (adjusted odds ratio), NA (not applicable), CI (confidence interval).

**Table 4 medicina-60-00826-t004:** Cumulative effect of association between socioeconomic risk factors and PTB in singleton births in Latvia during 2022.

			Model No. 4		Model No. 5	
	Crude OR (95% CI)	*p*-Value	aOR (95% CI) ^1^	*p*-Value	aOR (95% CI) ^2^	*p*-Value
Zero risk factors	Reference		Reference		Reference	
One risk factor	1.46 (1.26–1.70)	<0.001	1.34 (1.13–1.58)	<0.001	1.31 (1.11–1.55)	0.001
Two or three risk factors	2.29 (1.64–3.20)	<0.001	1.70 (1.15–2.51)	0.007	1.59 (1.07–2.35)	0.021

^1^ Risk groups, maternal comorbidities, maternal BMI before pregnancy, unhealthy habits of the mother, and maternal age were adjusted. ^2^ Risk groups, maternal comorbidities, maternal BMI before pregnancy, unhealthy habits of the mother, maternal age, pregnancy complications, and antenatal care usage were adjusted. OR (odds ratio); aOR (adjusted odds ratio), CI (confidence interval).

## Data Availability

The complete set of tables used during the current study is available from the corresponding author upon reasonable request. The MBR data have been given for this specific study, and the data cannot be shared without authorization from the register keepers.

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
