# Peer review of "Relationship between Maternal Socioeconomic Factors and Preterm Birth in Latvia"

_medicina, 2024, doi:10.3390/medicina60050826_

Round 1
Reviewer 1 Report
Comments and Suggestions for Authors
Review of manuscript submitted to Medicina-2944201:
Relationship between Maternal Socioeconomic Factors and 2 Preterm Birth in Latvia
By Kūkoja et al.
Very interesting study, well developed, analyzed and written. I have a few comments.
Abstract
Page 1
In Methods
Line 17 “A retrospective cross-sectional study” should be changed to “A cross-sectional study”
In the Results Section: Delete the following text [negative results not necessary to mention in the abstract]:
Lines 20-21: “While marital status and place of residence did not exhibit statistically significant associations with PTB,”
And lines 25-28: “Although nationality initially 25 showed significance, the association weakened and lost its statistical significance after adjusting for 26 other risk factors (aOR=1.16, p=0.085, CI 0.98-1.38), including unhealthy habits, maternal age, and 27 antenatal care service usage.”
Then keep these:
Lower maternal education was strongly connected with increased odds. Mothers with education levels be-22 low secondary education had over two times higher odds of PTB (aOR=2.07, p<0.001, CI 1.58-2.70) 23 and those with secondary or vocational secondary education had one and a half times higher odds 24 (aOR=1.58, p<0.001, CI 1.33-1.87) after adjusting for other risk factors.
And add, a short statement related to Table 3 results [page 7- on risk factor and two-three factors vs zero], on the cumulative effect socioeconomic risk factors and PTB.
Manuscript Text
Methods
Line 67 “A cross-sectional retrospective study” should be changed to “A cross-sectional study”
Line 94
It needs to specify the version of the IBM SPSS used (e.g., 27 or 28 or other)
Figure 1
Second row, ‘lower then secondary vs higher education’ a typo needs correction as ‘lower than secondary vs higher education.
Author Response
Dear Reviewer,
We sincerely appreciate your thorough review of our manuscript. Your insightful comments have greatly contributed to enhancing the quality of our work.
Please find point by point answers to your comments below:
Comment 1: Line 17 “A retrospective cross-sectional study” should be changed to “A cross-sectional study”
Answer: Thank you! Correction is available in Line 17
Comment 2: In the Results Section: Delete the following text [negative results not necessary to mention in the abstract]:
Lines 20-21: “While marital status and place of residence did not exhibit statistically significant associations with PTB,”
And lines 25-28: “Although nationality initially 25 showed significance, the association weakened and lost its statistical significance after adjusting for 26 other risk factors (aOR=1.16, p=0.085, CI 0.98-1.38), including unhealthy habits, maternal age, and 27 antenatal care service usage.”
Then keep these:
Lower maternal education was strongly connected with increased odds. Mothers with education levels be-22 low secondary education had over two times higher odds of PTB (aOR=2.07, p<0.001, CI 1.58-2.70) 23 and those with secondary or vocational secondary education had one and a half times higher odds 24 (aOR=1.58, p<0.001, CI 1.33-1.87) after adjusting for other risk factors.
And add, a short statement related to Table 3 results [page 7- on risk factor and two-three factors vs zero], on the cumulative effect socioeconomic risk factors and PTB.
Answer: Thank you. Corrections are available in Lines 20-27.
Comment 3: Line 67 “A cross-sectional retrospective study” should be changed to “A cross-sectional study”
Answer: Thank you! Correction is available in Line 75.
Comment 4: Line 94. It needs to specify the version of the IBM SPSS used (e.g., 27 or 28 or other)
Answer: Thank you. Correction is available in Line 115.
Comment 5: Second row, ‘lower then secondary vs higher education’ a typo needs correction as ‘lower than secondary vs higher education.
Answer: Thank you. Correction is available in Figure 1.
Please find attached the revised publication.
Thank you for your time and attention!

Reviewer 2 Report
Comments and Suggestions for Authors
The article presented to me for evaluation has been prepared carefully and in accordance with the requirements of the Medicina journal. The article raises an important topic regarding premature births and the factors influencing them. I have a few comments:
Whenever using an abbreviation for the first time, it should be expanded in the abstract, text and tables.
What new information does this study provide? what scientific gap does it fill?
Please describe the study group in more detail and provide the main characteristics
I suggest adding Flow-chart
I suggest putting the description from lines 117-136 in a table - it will be much more readable
On what basis was the level of education determined, was it referred to the ISCED classification?
Author Response
Dear Reviewer,
We sincerely appreciate your thorough review of our manuscript. Your insightful comments have greatly contributed to enhancing the quality of our work.
Please find point by point answers to your comments below:
Comment 1: Whenever using an abbreviation for the first time, it should be expanded in the abstract, text and tables.
Answer: Thank you! Corrections are available in Lines - 35, 75-76, 85, 108, 175, 194, 221, 260.
Comment 2: What new information does this study provide? what scientific gap does it fill?
Answer: Thank you! Information about relevance of the study can be found in Lines 71-73 and 375-379.
Comment 3: Please describe the study group in more detail and provide the main characteristics. I suggest adding Flow-chart.
Answer: Thank you! We have added extra table (Table 1) outlining the main baseline socioeconomic characteristic of the study participant, as well as additional information in the text. Please see Lines 78 and 85-94.
We would also like to stress that the study was restricted to all women with singleton pregnancies in Latvia during 2022. Please see Lines 78 and 85-94.
Comment 4: I suggest putting the description from lines 117-136 in a table - it will be much more readable.
Answer: Thank you! Corrections have been made. As suggested, part of the description from lines 117-136 were shortened in the text and integrated in Table 1 and part of it has been integrated in Table 2. Please see Lines 149-153.
Comments 5: On what basis was the level of education determined, was it referred to the ISCED classification?
Answer: The level of education was assessed using the structure outlined in the Medical Birth Register (MBR) of Latvia, which is based on Newborn cards (form 098/u). This framework classifies education levels into the following categories: no primary education, primary education, secondary education, secondary vocational education, higher education, and unfinished higher education. For the purposes of this study, these categories were further disaggregated as follows:
- Lower than secondary education (ISCED 0-2) - no primary education, primary education
- Secondary or vocational secondary education (ISCED 3-4) - secondary education, secondary vocational education, unfinished higher education
- Higher education (ISCED 5-8) - higher education
Thank you for outlining this question. To ensure the comparability of study results, we have provided the categorization of higher education according to ISCED within brackets in publication text. Please see Lines 102-103.
Please find attached the revised publication.
Thank you for your time and attention!

Reviewer 3 Report
Comments and Suggestions for Authors
This cross-sectional study analyzed the relationship between socioeconomic factors of pregnant women in Latvia and preterm birth, emphasizing the importance of maternal education in improving pregnancy outcomes. I have the following suggestions for the article:
1. Since the relationship between socioeconomic factors and preterm birth has been studied in other countries, it is recommended to introduce the uniqueness of Latvia's economy, demographics, and other factors compared to other countries in the introduction section to emphasize the necessity of this study in Latvia.
2. The results of this study are significant for formulating relevant policies to improve pregnancy outcomes. It is suggested to add a specific section in the discussion that demonstrates what specific measures government departments can take based on the study results and the expected outcomes.
3. Maternal education level may impact preterm birth through various pathways. It is recommended to increase the discussion on intermediary factors between the two.
4. It is advised to discuss whether the results of this study can be extrapolated to other regions outside of Latvia or other populations.
Author Response
Dear Reviewer,
We sincerely appreciate your thorough review of our manuscript. Your insightful comments have greatly contributed to enhancing the quality of our work.
Please find point by point answers to your comments below:
Comment 1: Since the relationship between socioeconomic factors and preterm birth has been studied in other countries, it is recommended to introduce the uniqueness of Latvia's economy, demographics, and other factors compared to other countries in the introduction section to emphasize the necessity of this study in Latvia.
Answer: Thank you! Information has been added. Please see Lines 62-73.
Comment 2: The results of this study are significant for formulating relevant policies to improve pregnancy outcomes. It is suggested to add a specific section in the discussion that demonstrates what specific measures government departments can take based on the study results and the expected outcomes.
Answer: Thank you! We have added two additional paragraphs in the discussion regarding this question. Please see Lines 288-301.
We would like to inform you, that separate publication will be developed with an aim to explore relevant policies to improve pregnancy outcomes in Latvia.
Comment 3: Maternal education level may impact preterm birth through various pathways. It is recommended to increase the discussion on intermediary factors between the two.
Answer: Thank you! We have added information in the discussion. Please see Lines 279-287.
Comment 4: It is advised to discuss whether the results of this study can be extrapolated to other regions outside of Latvia or other populations.
Answer: We have added additional paragraph in the discussion about how results of this study can be extrapolated to other regions outside of Latvia or other populations. Please see Lines 347-349.
Please find attached the revised publication!
Thank you for your time and attention!

Round 2
Reviewer 2 Report
Comments and Suggestions for Authors
Dear Authors,
thank you for all made corrections. You manuscript looks good.
best wishes
Reviewer